# Treating Critically Ill Patients Experiencing SARS-CoV-2 Severe Infection with Ig-M and Ig-A Enriched Ig-G Infusion

**DOI:** 10.3390/antibiotics10080930

**Published:** 2021-07-30

**Authors:** Alberto Corona, Giuseppe Richini, Sara Simoncini, Marta Zangrandi, Monica Biasini, Giuseppe Russo, Mauro Pasqua, Clemente Santorsola, Camilla Gregorini, Chiara Giordano

**Affiliations:** Accident & Emergency and Anaesthesia and Intensive Care Medicine Department, Esine and Edolo Hospitals, ASST Valcamonica, 25043 Brescia, Italy; giuseppe.richini@asst-valcamonica.it (G.R.); sara.simoncini@asst-valcamonica.it (S.S.); marta.zangrandi@asst-valcamonica.it (M.Z.); monica.biasini@asst-valcamonica.it (M.B.); giuseppe.russo@asst-valcamonica.it (G.R.); mauro.pasqua@asst-valcamonica.it (M.P.); clemente.santorsola@asst-valcamonica.it (C.S.); cam.gregorini@gmail.com (C.G.); c.giordano003@unibs.it (C.G.)

**Keywords:** COVID-19, critically ill patients, secondary bacterial infection, co-infection

## Abstract

SARS-CoV-2 in patients who need intensive care unit (ICU) is associated with a mortality rate ranging from 10 to 40–45%, with an increase in morbidity and mortality in presence of sepsis. We hypothesized that IgM and IgA enriched immunoglobulin G may support the sepsis-related phase improving patient outcome. We conducted a retrospective case–control study on 47 consecutive patients admitted to our ICU. At the time of admission, patients received anticoagulants (heparin sodium) together with the standard supportive treatment. We decided to add IgM and IgA enriched immunoglobulin G to the standard therapy. Patients receiving IgM and IgA enriched immunoglobulin G were compared with patients with similar baseline characteristics and treatment, receiving only standard therapy. The mortality resulted significantly higher in patients treated with standard therapy only (56.5 vs. 37.5%, *p* < 0.01) and, at day 7, the probability of dying was 3 times higher in this group. Variable life adjustment display (VLAD) was 2.4 and −2.2 (in terms of lives saved in relation with those expected and derived from Simplified Acute Physiology Score II) in the treated and not treated group, respectively. The treatment based on IgM and IgA enriched immunoglobulin G infusion seems to give an advantage on survival in SARS-CoV-2 severe infection.

## 1. Introduction

In 2019, a new β-coronavirus named Severe Acute Respiratory Syndrome Coronavirus 2 (SARS-CoV-2) emerged in December in Wuhan City, Hubei Province, China, that caused a pandemic and an unprecedented global crisis. Despite rigorous global containment and quarantine efforts, the outbreak spread and reached the rest of the world, with 162.177.376 cases and 3.364.178 deaths worldwide (as of 17 May 2021) [1] and a case fatality rate ranging from 0.2% to 29% depending on demographics and national health systems [2]. The virus is transmitted by the respiratory system via droplets and aerosols, and in most cases causes mild respiratory symptoms or remains asymptomatic [3].

In approximately 5% of patients, unclear molecular mechanisms trigger the onset of an immune imbalance, which causes a rapid progression to either acute respiratory distress syndrome (ARDS) and to multiple organ dysfunction syndrome (MODS) [4,5]. SARS-CoV-2 infection is associated with (i) hyper-inflammation [6]; (ii) changes in cytokine production (‘cytokine storm’) (IL-1β, IL-1ra, IL-6, IL-10, TNF-α, GM-CSF, IL-17); (iii) pathological shifts of circulating leukocyte subsets [7,8].

These severe conditions lead to the activation of a sepsis-like inflammation (pro-inflammatory phase of sepsis/SIRS) and an impairment of the protective immunity against viral infection (immunoparalysis phase of sepsis/SIRS). The latter, which worsens the virus-induced lymphopenia, may result in the failure of the adaptive immune system in developing functional immunoglobulins and in clearing pathogens [9]. Pulmonary or cardiovascular complications and coagulopathy might also occur [10,11]. ICU admission is generally associated with a variable mortality rate (8–38%) [12], that reaches 40–45% in case of embolism and multiple organ failures (renal, hepatic) [12].

In theory, the use of IgM and IgA enriched immunoglobulin G in patients showing signs of both hyper- and hypo-inflammation could represent an effective therapeutic strategy [13]. Unfortunately, to date, only one case report has been published on the beneficial use of this treatment for SARS-CoV-2 infections [14]. Investigations using IgM- and IgA-enriched immunoglobulin are still on the way.

For this reason, we hypothesized that IgM and IgA enriched immunoglobulin G may help and support both sepsis-related phases and microcirculation [15], improving patient outcome despite the side effects caused by this therapy [13]. Therefore, we conducted a retrospective observational study on patients treated with IgM and IgA enriched immunoglobulin G in addition to the standard therapy admitted to our ICU after clinical worsening in the ward.

## 2. Results

During the study period, 2473 patients were admitted to the Accident and Emergency Department, 710 of which were hospitalized and 51 considered severely ill. In total, 47 patients were admitted to our ICU and four of them transferred to a tertiary referring center due to needing Extracorporeal Membrane Oxygenation (ECMO) respiratory support.

Group A included 24 patients treated with IgM and IgA enriched immunoglobulin G in addition to the standard therapy while Group B involved 23 patients that received the standard therapy only. Remdesivir and tocilizumab were used only for 2 patients (remdesivir in a patient of Group B and tocilizumab in a patient of Group A). Table 1, Table 2 and Table 3 report baseline characteristics of patients, comorbidities and therapy administered before transfer to ICU. There was no significant difference or values above the higher limit in the parameters recorded after IgM/IgA enriched IgG treatment between Groups A and B, although the median (IQR) was higher in the treated group (see Appendix A). Table 4 reports the pathological events that occurred during the ICU stay. There were no significant differences in the baseline characteristics between the two groups, except for higher mean level of CPK (2239 in Group A vs. 3467 in Group B) and use of corticosteroids in Group B (12.5% vs. 43.5%, *p* = 0.043). The treatment received by patients during the ICU stay was not statistically different (Appendix A). The analysis of the pathological events showed that the rate of acute kidney failure was higher in Group A than in Group B (10 vs. 3 patients, OR: 5.15 95%CI = 1.2–23, *p* = 0.027). The infection rate, the percentage of septic shock and the death rate are also reported. The whole profile of the patient is shown in the online supplement (Appendix A).

### Mortality

The recorded overall mortality was significantly higher in Group B (56.5 vs. 37.5%, *p* = 0.01) (Table 4). Figure 1A shows Kaplan–Meier analysis assessing the death likelihood in relation with ICU length of stay and stratified according to groups (A and B) of treatment. The intriguing data is that on day 7 following ICU admission, patients not treated with IgM and IgA enriched IgG had a death likelihood 3 times higher, although not statistically significant (*p* > 0.05).

VLAD showed a higher number of saved lives in Group A (+2.4) when compared to Group B (−2.2) in relation with expected death likelihood associated to SAPS II (Figure 1B).

Appendix A show the same categories and quantitative variables of Appendix A but further stratified according to the outcome.

## 3. Discussion

The initial period of the pandemic was marked by an overwhelming patient admission to the Italian hospitals, particularly in Lombardia. Worldwide, all health systems are participating in the effort of increasing their capacity to sustain the growing number of COVID-19 patients. At the beginning of the outbreak, our unit had six beds, but the number of beds increased to 17, plus four high dependence beds and over 50 ward beds used for NIPPV or c-PAP, throughout the pandemic. The supply of personal protective equipment (PPE) and drugs was also critical, due to the increasing use everywhere in Italy, Europe and around the world. Drawing and performing randomized comparative studies is particularly difficult and often unrealistic.

Several therapies have been tested for the treatment of SARS-CoV-2 infection, in particular for the severe form of pneumonia that leads to ARDS. The pathophysiology of the disease, unknown at the beginning of the pandemic, has received growing contributions, but a clear consensus exists for few treatments only [16].

Moreover, the associated SARS-CoV-2 sepsis can cause a catastrophic increase in morbidity and mortality [17].

The clinical picture of patients affected by COVID-19 who develop sepsis is particularly severe and characterized by a wide range of signs and symptoms of multiorgan involvement, with alteration of almost all laboratory parameters. Respiratory manifestations (dyspnea and hypoxemia), renal failure, tachycardia, coagulopathy and altered state of consciousness are generally observed. The SOFA score, based on the results of laboratory tests and clinical data, is the parameter that has mostly demonstrated a prognostic value in predicting mortality within the ICU [18].

There is no robust evidence supporting the use of IgM enriched IVIG in patients with COVID-19. A number of descriptive observational studies have been carried out with unclear results [19,20,21,22].

Moreover, there is insufficient evidence to support the use of convalescent plasma or hyper-immune immunoglobulin isolated from the blood of patients who have recovered from COVID-19 [23].

IgM and IgA enriched immunoglobulin G has been successfully used for many years in the prophylaxis and treatment of severe sepsis and septic shock [13,24,25,26,27], particularly caused by Gram-negative bacteria [28]. A recent meta-analysis reported a reduction in mortality and ventilation time with the use of IgM and IgA enriched immunoglobulin G in patients with sepsis or septic shock compared with the respective control groups (relative risk (RR) 0.60) [25]. IgM and IgA enriched immunoglobulin G use in patients undergoing hematopoietic stem cell transplantation (HSCT) produced a significant decrease in infection-associated transplant related mortality rate [29], decreased the key inflammasome IL-1β molecule in an *Escherichia coli*-model of pig’s sepsis [30] and reduced the hepatic toxicity in patients who underwent bone marrow transplant [31]. Additionally, this treatment has been used successfully in SARS patients unresponsive to corticosteroids and ribavirin, with significant improvement in radiographic score and oxygen requirement [32,33].

Recently, Carannante has reported a case of early administration of IgM and IgA enriched immunoglobulin G, which has been effective in a patient with SARS-CoV-2 infection, allowing clinical remission and improvements in radiological findings in a few days [14].

These considerations led us to analyze a cohort of ICU patients affected by COVID-19 treated with IgM and IgA enriched immunoglobulin G in comparison with no IgM/IgA enriched IgG treated patients.

In our retrospective observational study on patients affected by severe COVID-19 pneumonia and sepsis admitted to ICU, the treatment with IgM and IgA enriched IgG infusion in addition to the standard therapy resulted in a higher survival rate compared to standard therapy only. It has been reported that IgM and IgA enriched immunoglobulin G had a positive impact on coagulation processes, decreasing endotoxin activity while increasing PLT number and fibrinogen [34], that resulted in a positive impact on microcirculation and perfusion in patients with sepsis [15]. Considering the increasing amount of data about the pathophysiology of COVID-19 and the key role of coagulation and microcirculation alterations [35,36,37], this could be a good rationale to support the beneficial effect of an IgM and IgA enriched immunoglobulin G therapy.

In patients at risk for acute renal failure, IVIg products should be administered at the minimum rate of infusion and dose practicable [38]. The lower rate of acute renal failure observed in the untreated group could be explained by the protective effect of the corticosteroids (as also reported by other authors [39,40,41,42]) frequently administered to these patients before the admission to ICU; moreover, the longer length of survival in the treated group could have allowed the onset of this complication. In fact, a higher survival rate has been observed in Group A patients which translates into a longer ICU stay and mechanical ventilation need, whose likely interpretation is to be referred to a better control of SARS-CoV-2 hyper-inflammation and immunoparalysis after IgM and IgA enriched immunoglobulin G infusion [13,14,15]. A significantly longer period of ICU stay may also justify the higher likelihood of organ failure as complication of ICU longer treatment, as shown in Table 4. However, most patients developed acute kidney (8/10, 80%) and liver (8/10, 80%) failure at the time of ICU admission, before the infusion of IgM and IgA enriched immunoglobulin G whose posology was adjusted anyway on the basis of kidney failure [13].

Interestingly, these patients developed a lower (Table 4) rate of sepsis and septic shock. This might be a consequence of the IgM and IgA enriched immunoglobulin G treatment that helps and supports the resolution of complications due to a secondary bacterial sepsis in relation to their adjuvant support to the sepsis-related phases and the microcirculation, impaired in the severe stage of infection [13,14,15].

Certainly, if compared with specific intravenous IgG, formulations containing IgG, IgA and IgM exert a significant effect on mortality in patients with sepsis and septic shock. Polyvalent Ig plays a role in (i) recognizing, opsonizing and eliminating pathogens, (ii) neutralizing the exotoxins; moreover, such preparations contain antibodies directed against the microorganism lipopolysaccharide and remove and inhibit the gene transcription of inflammatory mediators [10]. It is probable that it is thanks to their pentameric structure that IgM can prove superior to IgG in neutralizing toxins and in microorganism agglutination. It has been shown that the bactericidal effect is more important with regard to IgM, since these are able to activate complement about 400 times more than IgG, as well as about 1000 times more effective in microorganisms opsonization [43].

Our study has some limitations such as the observational retrospective design on a low number of patients, which does not allow to draw definite conclusions. However, as far as we know, it is the first case series of COVID-19 patients treated with IgM and IgA enriched immunoglobulin G. Indeed, the differences recorded in terms of death likelihood on day 7 and above all, between expected and recorded mortality (VLAD) may provide an intriguing suggestion on the management of SARS-CoV-2 related SIRS.

In conclusion, our observational study can contribute to the small amount of data available on this matter and our promising results may be a rationale for wider, randomized studies.

## 4. Materials and Methods

The study was conducted in accordance with the Declaration of Helsinki and informed consent for treatment and data collection was obtained from every patient or respective legal representatives.

Our retrospective study was conducted on 47 consecutive patients admitted to our ICU from 1 March 2020, up to 15 April 2020; before the ICU admission, patients received standard therapy with hydroxychloroquine, lopinavir-ritonavir, steroids, azithromycin. After the ICU admission, these drugs were withdrawn, and the patients received therapy/prophylaxis of pulmonary embolism with ASA and heparin sodium. After the publication of data about the utility of using immunoglobulins in patients with severe COVID-19 infections [14], we therefore hypothesized that these patients could benefit from IgM and IgA enriched immunoglobulin G. From 4 March 2020, patients receiving the immunoglobulin treatment in addition to the standard therapy were considered as part of Group A, whereas patients treated before this date receiving only the standard therapy were considered as part of Group B. All the data of the patients were mandatorily recorded in an interactive electronic database, from which some data was extracted to compare the outcome of treatments in the two groups of patients.

From the electronic database, we extracted the data of 47 consecutive patients admitted into ICU for respiratory failure due to SARS-CoV-2.

Among the mandatory data recorded in our database, we analyzed the following baseline data (Table 1):age, history of the disease, including the length of stay (LOS), diagnostic and therapeutic procedure received until transfer to the ICU;clinical characteristics of the patients, including laboratory tests;comorbidities like hypertension, heart disease, COPD, neoplasms, diabetes, chronic renal failure, vascular tree diseases, chronic pathological or iatrogenic immunosuppression;sepsis-related Organ Failure Assessment (SOFA) score and Simplified Acute Physiology Score II (SAPS II);previous non-invasive positive pressure ventilation (NIPPV);the following parameters/events recorded during the staying in ICU were analyzed for our study (Table 2). Other parameters are reported in Appendix A;death;severe pulmonary embolization;continuous renal replacement therapy (CRRT) and its duration;echocardiographic signs of myocardial failure;hepatic insufficiency and alteration of hepatic cytolysis and cholestatic indices;variable life adjustment display (VLAD), a type of indicator used to measure healthcare quality and patient outcomes. VLAD allows to ascertain the difference between expected mortality (indicated by SMR (standardized mortality rate) derived from SAPS II and recorded mortality) (for the explanations see Appendix A) [43,44].

### 4.1. Therapy

From the beginning of our observation date (1 March 2020) the patients received a standard therapy with anticoagulants (heparin sodium) at the time of ICU admission (Group B); after the date of 4 March 2020, they received the addition of IgM and IgA enriched immunoglobulin G (Group A). IgM and IgA enriched immunoglobulin G were administered at 500 mg/kg/day and infused over 48 h. The preparation consists of a mix of 76% of IgG, 12% of IgM and 12% of IgA and, in more detail, of (i) 380 mg/kg/day of IgG, (ii) 60 mg/kg/day of IgM and (iii) 60 mg/kg/day of IgA. In case of severe renal failure (creatinine clearance <30 mL/min), half of the dose was administered following our protocol. In the Appendix A, the applied algorithm for the corrected posology is shown (Appendix A). The standard therapy remained the same for the two groups.

### 4.2. Primary Outcomes

The primary outcomes of this study were the mortality rate adjusted according to the VLAD score.

### 4.3. Secondary Outcomes

Infection rate and septic shock;Occurrence of severe pathological complications.

### 4.4. Statistical Analysis

For quantitative discrete variables Student’s T test was used, with the Mann–Whitney’s non-parametric U test correction, in case of non-normal distribution; for categorical variables the Chi-square test was applied to determine differences between proportions with Fisher correction in the case of expected values <5. The univariate odd ratio and statistical significance were identified with Mantell–Hentschel test.

### 4.5. Mortality

Kaplan–Meier method was used to assess the risk of mortality stratified between the two groups of the study; the length of survival was calculated from the onset of SARS-COV-2 infection symptoms up to death or hospital discharge. Moreover, the standardized mortality rate (SMR) was computed from SAPS II to get the “attended mortality” for each patient. The Variable Life Adjusted Display (VLAD) was used to assess differences between attended mortality and recorded mortality (A, B).

Statistical significance: we assumed that the α error is ≤0.05 associated with a power of the tests not ≤0.9.

## Figures and Tables

**Figure 1 antibiotics-10-00930-f001:**
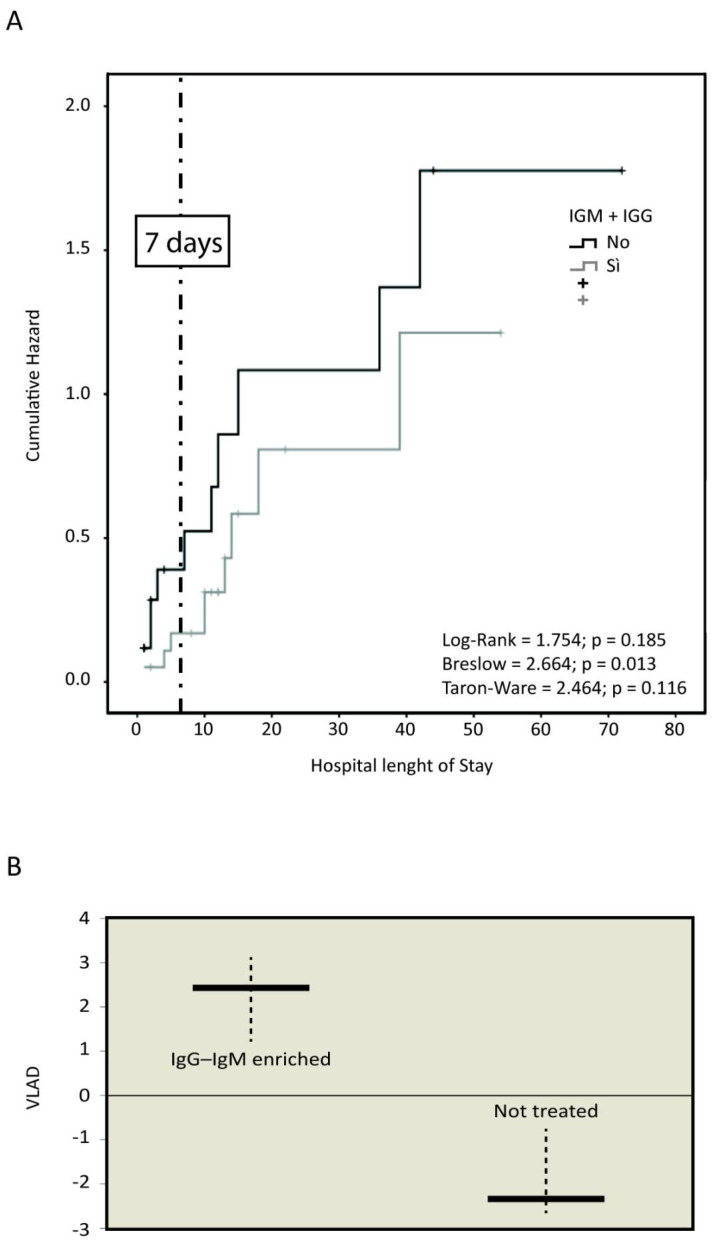
Mortality. (**A**) Recorded death likelihood on day 7 since the ICU admission. (**B**) Comparison of VLAD in relation to expected death likelihood.

**Table 1 antibiotics-10-00930-t001:** Baseline characteristics of patients enrolled.

Variables	Group A *n* = 24	Group B *n* = 23	*p*-Value
Age (yrs.)	59.5 (57–66)	61(55–67)	0.212
SAPS II-score	45 (38–59)	46 (38–58)	0.222
SOFA-score	9 (7–10)	8 (5–11)	0.174
Pre-ICU-LOS (days)	3 (1–5)	4 (1–10.5)	0.620
ICU-LOS (days)	15.5 (5–15.5)	9 (5–11)	0.045
MV-length (days)	12 (4–15)	7 (5–9)	0.045
Urea (mg/dL)	55 (40–109)	48 (30–74)	0.209
Total bilirubin (mg/dL)	0.9 (0.6–1.9)	1 (0.8–1.7)	0.529
Ammonium (mg/dL)	76 (57–109)	72 (53–100)	0.335
Total diuresis (mg/dL)	1.7 (1.2–2.2)	1.7 (1.1–2.6)	0.202
PLT (103/mL)	110 (75–275)	105 (86–233)	0.358
LDH (mg/dL)	525 (320–705)	462 (272–563)	0.478
CPK (mg/dL)	2239 (1140–2537)	3467 (2228–4185)	0.022
D-Dimer μg/mL	6485 (1400–11.000)	10.7 (1.200–20.000)	0.110
IgG-pre (mg/dL) (range 7.5–15.6)	10 (7–14.5)	10.6 (6.8–14.6)	0.233
IgA-pre (mg/dL) (range 0.82–4.53)	0.8 (0.7–3.1)	0.9 (0.8–3)	0.125
IgM-pre (mg/dL) (range 0.46–3.04)	0.5 (0.4–1.5)	0.6 (0.4–1.6)	0.276

SAPS II: Simplified Acute Physiology Score II; SOFA: Sepsis-related Organ Failure Assessment; LOS: length of stay.

**Table 2 antibiotics-10-00930-t002:** Comorbidities of patients enrolled.

Variables	Group A *n* = 24	Group B *n* = 23	OR (95% CI)	*p*-Value
IDDM/NIDDM	2 (8.3%)	2 (8.7%)	1 (0.2–7.9)	1
Diabetes	2 (8.3%)	2 (8.7%)	1 (0.2–7.9)	1
Smoke	2 (8.3%)	0	n.a.	0.488
COPD and other lung pathologies	2 (8.3%)	4 (17.4%)	0.4 (0.1–2.6)	0.416
Chronic renal failure	1 (4.2%)	0	n.a.	1
Chronic liver failure	0	0	n.a.	n.a.
Steroid therapy and immunosuppression	0	2 (8.7%)	n.a.	0.234
Hypertension	12 (50%)	7 (33.3%)		
CAD	4(16.7%)	1 (4.3%)	4.7 (0.5–6.2)	0.343
Other cardiopathies	2 (8.3%)	2 (8.7%)	0.9 (0.1–7.4)	1
Cerebral stroke	1 (4.2%)	0	n.a.	1
Neoplasia	1 (4.2%)	0	n.a.	1

CAD: coronary artery disease.

**Table 3 antibiotics-10-00930-t003:** Therapy before ICU admission.

Treatment	Group A *n* = 23	Group B *n* = 24	OR (95% CI)	*p*-Value
Antiviral therapy (*)				
Hydroxychloroquine	7 (29.2%)	10 (43.5%)	0.6 (0.2–1.9)	0.530
Lopinavir/ritonavir	6 (25%)	5 (21.7%)	1.3 (0.3–5.1)	1
Steroids	3 (12.5%)	10 (43.5%)	0.2 (0.1–0.9)	0.043
Azitromycin	7 (29.2%)	6 (26.1%)	1.3 (0.3–4.6)	1

* Withdrawn at the ICU admission.

**Table 4 antibiotics-10-00930-t004:** Complications during ICU stay.

Event	Group A *n* = 24	Group B *n* = 23	OR (95% CI)	*p*-Value
Pulmonary embolism	3 (12.5%)	6 (26.1%)	0.4 (0.2–1.8)	0.277
Acute kidney failure	10 (41.7%)	3 (13%)	5.1 (1.2–23)	0.043
CRRT	4 (16.7%)	1 (4.3%)	4.9 (0.5–48.4)	0.185
Acute liver failure	10 (41.7%)	5 (21.7)	2.7 (0.8–10.2)	0.197
Acute cerebral stroke	2 (8.3%)	2 (8.7%)	0.9 (0.1–7.5)	1
Sepsis	5 (20.8%)	9 (39.1%)	0.6 (0.1–1.3)	0.116
Septic shock	3 (12.5%)	7 (21.7%)	0.7 (0.2–5)	0.045
Deaths	9 (37.5%)	13 (56.5%)	0.6 (0.2–0.8)	0.01

CRRT: continuous renal replacement therapy.

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
