# Peer review of "Treating Critically Ill Patients Experiencing SARS-CoV-2 Severe Infection with Ig-M and Ig-A Enriched Ig-G Infusion"

_antibiotics, 2021, doi:10.3390/antibiotics10080930_

Round 1

Reviewer 1 Report

The authors did a retrospective study on covid patients receiving standard treatment vs groups who received additional IgA and IgM antibody infusion and state that these patients had a better outcome. However, the  analysis and especially the results section is not self-explanatory and needs to be greatly improved.

Reviewer 2 Report

The paper states that IgM & IgA-rich IVIG administration significantly increased survival rates in SARS-CoV-2 patients requiring intensive care (ICU). It is intended to convey useful information for the treatment of ICU in COVID-19, but detailed information on the drug to be administered is unknown, and it is unclear whether it is equivalent to the commonly used IVIG.

Therefore, the authors need to clearly reconsider the relationship between the information on the drug used for treatment and the duration of administration and recovery.

The comments are described below.

Introduction

  • Clarify the rationale for why IgM and IgA-rich immunoglobulins have been hypothesized to be effective in treating COVID-19.

Result

  • The patient grouping (A & B) is described in the M & M, but for the reader's understanding, please state the rationale for the grouping in the text and describe it in an easy-to-understand manner.
  • Differences between groups based on IgM and IgA dose performance should be clearly shown. I think it is also useful information regarding the total dose until the cure is judged. Please also add a reference to side effects.
  • “Remdesivir” and “Tocilizumab” administration differ between groups. Add an explanation of the differences in medication choices and the relationship between these differences in mortality in the results and discussion.
  • Specify the number of patient virus copies at the start of IVIG treatment. Since it is caused by a viral infection, it is important to show the relationship with the viral load before and after the start of treatment.
  • Add a description of the legend in Figure 1.

Discussion

Materials and methods

  • Do you have an approval number for ethical approval matters in the investigation?
  • Detailed information on the formulation should be added. IVIG is rich in IgG, but specify how much IgM and IgA it contains.
  • It should be clearly stated whether there was a difference between the groups on the day of illness when they were transported to the ICU and started IVIG administration. Add to the discussion how the relationship with complications is related to the therapeutic effect of IVIG.

Reviewer 3 Report

The condition of COVID patients who need intensive care is associated with a mortality ranging from 10 to 40-45%, with increase in morbidity and mortality in case of sepsis. The authors hypothesized that IgM and IgA enriched immunoglobulin transfusion (IVIG) could support the sepsis-related phase to improve patient outcome, and they examined the hypothesis with a retrospective case-control study on 47 consecutive patients admitted to ICU. Patients receiving IgM and IgA enriched immunoglobulin were compared with patients with similar baseline characteristics but received only standard therapy. The authors found that the mortality resulted significantly higher in patients with standard therapy (56.5 vs. 37.5%, p<0.01) which suggested the beneficial effect of IgM and IgA enriched IVIG treatment on survival of patients with SARS-CoV-2 severe infection.

In this manuscript, the authors provided comparison of clinical outcomes of patients of severe COVID admitted to ICU, with or without IgM/IgA-enriched IVIG therapy. Despite the cohort size is small, most of the baseline data are well balanced between the IVIG-treated and -untreated group. Additionally, the manuscript will add information in a topic that is currently understudied.

Below are my comments for the authors to consider:

  1. Line 29: COVID-19 refers to the disease not the name of the virus. SARS-CoV2 is the virus.
  2. Line 152-153: the authors stated that the patients in IVIG treated group have longer length of survival. Please clarify how was the length of survival determined? Was it from the time of diagnosis for COVID to the time of death for deceased patients? What about the patients that stayed alive during the study period? Can the authors provide this part of information in the manuscript?
  3. In the result section, starting line 55, it will be better to describe clearly that Group A and Group B are assigned from the 47 ICU patients based on whether they received IgM/IgA-enriched IVIG. Can also refer to the method section for details.
  4. Fig 1A, labels within the graph should be in English.
  5. Fig 1B, should add axis label.
  6. Global data have shown that COVID severity and mortality rates may defer between male and female patients and patients of different ethnicity. Please provide information of gender and ethnicity proportion in this study.
  7. Please provide the regimen of the IVIG in more detail.
  8. Do the authors have any thoughts about the ICU-LOS being longer for patients admitted since 4/3/2020 (group A)? Was it the result of higher percentage of acute renal failure related to the IVIG treatment?
  9. Can the authors clarify on the variable “Infections” in table 4? Does it mean whether the patient developed sepsis during ICU stay? Or it refers to any bacterial infection detected during ICU stay?
  10. Also in table 4, do the authors have any thoughts about the higher ratio of patients having infections during ICU stay that got septic shock in IVIG-treated group? (Since the 5 patients that were infected in group A all have septic shock, whereas in group B, 7 out of the 9 patients infected during ICU stay underwent septic shock.)
  11. Each quantitative variable of the clinical data should clearly state its unit, both in main article and in supplemental tables. For example, “Age” should be shown as “Age (year)”, “Urea” should be shown as “Urea (mg/dL)” or Urea “(mmol/L)”. And full name for all abbreviations should be provided in the legend or the bottom of the table, or if in the main text, it should be spelled out when mentioned for the first time.

Round 2

Reviewer 1 Report

The authors can include the explanations they have provided about the figures and results in the response to reviewers in the main manuscript. 

Author Response

REVIEWER 1 _ ROUND 3

Point 1: The authors can include the explanations they have provided about the figures and results in the response to reviewers in the main manuscript. 

MADE MODIFICATIONS AS REQUESTED

Reviewer 2 Report

The revised manuscript now clearly shows the author's opinion. It is necessary to consider the following several points regarding the content of the manuscript. 

When correcting a comment answer, you should reply with an accurate indication of the position of the correction.

Major points:

In this paper, IgM & IgA enriched immunoglobulin is described effectively. Is it expected that the essential therapeutic effect will be achieved even if IgM & IgA is not included? Or do you recommend using it with emphasis on IgM & IgA-containing globulin? Is it possible to add to the discussion section on this point, including the selection when IgM & IgA enriched immunoglobulin is not available, for example?

In the title, it is described as IgM enriched immunoglobulin, but in the text, IgM & IgA enriched immunoglobulin is repeatedly expressed. Do the authors want to intend that only IgM contamination is effective as a globulin preparation for the use of this preparation? In relation to the above comment, please organize the notation of the title and the text.

Minor points:

The dates listed in L205 and 239 are unified, such as "April 15th, 2020".

Regarding the description of the cited references, it is necessary to revise the description in a manner according to the guide of the journal.

Reviewer 3 Report

The authors have made corrections and provided additional information to improve the quality of this manuscript. I have no further comment except for the reminder to please make sure all text is in English. (Section 4.1, currently shows “mg/kg/die”)

Author Response

REVIEWER 3 _ ROUND 3

The authors have made corrections and provided additional information to improve the quality of this manuscript. I have no further comment except for the reminder to please make sure all text is in English. (Section 4.1, currently shows “mg/kg/die”)

MADE CORRECTIONS